# Consumer Perception of Innovative Fruit and Cereal Bars—Current and Future Perspectives

**DOI:** 10.3390/nu16111606

**Published:** 2024-05-24

**Authors:** Małgorzata Kosicka-Gębska, Marta Sajdakowska, Marzena Jeżewska-Zychowicz, Jerzy Gębski, Krystyna Gutkowska

**Affiliations:** Institute of Human Nutrition Sciences, Warsaw University of Life Sciences (SGGW-WULS), Nowoursynowska 159 C, 02-776 Warsaw, Poland; marta_sajdakowska@sggw.edu.pl (M.S.); marzena_jezewska_zychowicz@sggw.edu.pl (M.J.-Z.); jerzy_gebski@sggw.edu.pl (J.G.); krystyna_gutkowska@sggw.edu.pl (K.G.)

**Keywords:** fruit and cereal bars, innovations, consumer needs and expectations

## Abstract

The aim of the study was to ascertain consumers’ interest in innovative fruit and cereal bars and their expectations of changes that could be applied by manufacturers to improve their health-promoting properties. An additional aim was to assess how these interests and expectations, as well as the information provided on the product label, determine the willingness to purchase a fruit and cereal bar with health-promoting properties. Data were collected through a quantitative CAWI survey conducted in 2020. It involved 1034 respondents. A logistic regression model was developed in which the dependent variable was the respondents’ interest in an innovative fruit and cereal bar on whose packaging the manufacturer declared its health-promoting properties. It was found that producers’ efforts to change the packaging to an organic one (OR = 1.24) along with enriching the product with chia seeds/flaxseed (OR = 1.22), vitamins and minerals (OR = 1.19), as well as fruit (OR = 1.14) and protein (OR = 1.12), or removing ingredients that cause allergies, would significantly increase the chance of respondents purchasing such a bar. A celebrity image and a claim that the product “helps maintain a healthy body weight” on the label would also encourage purchases. On the other hand, reducing the sugar content or enriching a fruit and cereal bar with powdered insects would significantly reduce the propensity to buy it.

## 1. Introduction

Fruit and cereal bars, like cereal bars, are products whose consumption has increased worldwide in recent years [1,2,3]. They are consumed by people on a diet, those involved in sports, people with health problems, or consumers looking to satisfy hunger and provide energy quickly [4,5]. The interest in such products is linked to changing lifestyles of consumers, who are increasingly interested in choosing foods that are convenient and yet functional. (Research shows that, in recent years, consumers have recognised the need to replace conventional snacks (e.g., chocolates, biscuits, and crisps) with fruit, dairy products, and various types of bars rich in health-beneficial ingredients [6].

Such bars may be a healthier and lower-calorie alternative to chocolate bars [7,8]. The enrichment of bars with nuts, fruits, and cereals promotes their acceptance due to their bioactive content [9,10,11]. Studies indicate that their consumption is also determined by respondents’ awareness of the fact that they provide more fibre [12,13]. Like sweets, these products improve mood. Therefore, consumers value them for their sweet taste, associated with safety and childhood [14].

Despite the above-mentioned positive characteristics, cereal and fruit and cereal bars, due to the fact that they belong to the category of sweets, cannot be considered products with unequivocally positive health effects [15]. Previous research confirms that excessive consumption of sweets contributes to a number of diseases, including emotional problems [16,17]. Research suggests that commercially available cereal bars are characterised by varying nutritional value [18,19]; among other things, they may be high in sugar and saturated fat [20]. From this perspective, these products are perceived as highly processed, with high amounts of additives, and thus as unnatural products [21]. And yet, consumers are increasingly concerned about their health, expecting the food they consume to be natural, health-promoting, and sustainable [22]. Such expectations are also held for fruit and cereal bars, prompting manufacturers to improve them [23]. Research is therefore needed to determine what innovations in cereal bars consumers may expect [7]. 

To date, research primarily related to cereal bars has focused on their production technology, with a particular focus on innovative additives [4,24]. There is little work investigating consumer acceptance of innovative fruit and cereal bars, or the changes expected by consumers that manufacturers could apply to such bars to improve their health-promoting properties. In response to these questions, a study was carried out to investigate (1) consumers’ interest in innovative fruit and cereal bars and (2) their expectations of changes that could be applied by manufacturers of fruit and cereal bars to improve their health-promoting properties, and then to evaluate how these interests and expectations, as well as the information provided on the product label, determine interest in purchasing an innovative fruit and cereal bar with health-promoting properties.

## 2. Materials and Methods

### 2.1. Ethical Approval

The Ethics Committee of the Faculty of Human Nutrition and Consumer Science, Warsaw University of Life Sciences (SGGW), appointed based on Regulation No. 27 of the SGGW Rector of 5 May 2016, approved the protocol of analysis of the behaviour of Polish consumers in the sweets market and of the determinants of consumer acceptance of innovative changes aimed at counteracting obesity, 25 June 2018, Resolution No. 30/2018, as consistent with the guidelines laid down in the Declaration of Helsinki. Informed consent was provided by participants.

### 2.2. Data Collection Process

A quantitative study of Polish consumers’ behaviour towards fruit and cereal bars was carried out in 2020 by a professional research agency, ARC Rynek i Opinia from Poland. Respondents were recruited from an online panel (ePanel) of 65,000 people. Due to the lack of national statistical data providing information on the level of consumption of cereal bars or fruit and cereal bars by Poles, respondents for the study were selected based on a purposive sampling method. No quotas were imposed on individual demographic variables (gender, age, education, place of residence). The study also collected basic demographic data on those who did not consume fruit and cereal bars. These subjects were excluded from the sample in the next research stage. Analysis of the data, including those consumers of this product category, did not reveal any significant deviations from the population profile of the Polish population in the selected demographic sets. After the survey questionnaire was developed by the authors of the article and a pilot study was conducted (35 completed questionnaires), the proper survey was carried out. Ultimately, 1034 adult respondents who consumed fruit and cereal bars and were 18 years of age or older participated in the study. A CAWI (computer-assisted web interview) method was used to collect data.

### 2.3. Questionnaire

The questionnaire used in the study covered issues relating to consumer behaviour towards fruit and cereal bars. 

To assess the interest in purchasing an innovative fruit and cereal bar, respondents were asked to select one of the following statements: 1/I want to try it out and, if possible, I buy it immediately without thinking about it; 2/it arouses my interest, I think about whether I will buy it, gather information, compare it with others and finally buy it; 3/I accept every novelty with distrust, I think about it for a long time and I only buy it when my friends have done so and are satisfied with the purchase; 4/new/innovative products generally do not interest me, I do not buy untested things.

Additionally, the respondents’ interest in purchasing an innovative fruit and cereal bar, on the packaging of which the manufacturer declared its health-promoting properties, was assessed. Respondents expressed their interest in such a product by choosing one of two answers: yes—I am interested in buying or no—I am not interested in buying.

The knowledge of innovations in fruit and cereal bars was assessed in relation to the following actions: 1/change in product weight, 2/change in packaging appearance, 3/change in packaging size, 4/change in packaging to organic, 5/enrichment of the product with new ingredients, 6/reduction in the level of ingredients unfavourable to health, 7/improvement of packaging to improve the shelf life of the product, 8/introduction of a product with a new flavour, 9/change in product composition, 10/introduction of new ways of food preservation, 11/introduction of changes contributing to environmental protection, 12/the manufacturer proposing new uses for the product. Opinions concerning each measure were presented on a 5-point scale, with a score of 1 indicating “totally disagree”; 2, “disagree”; 3, “neither agree nor disagree”; 4, “agree”; and a score of 5 indicating “totally agree”.

Respondents’ expectations of changes that could be applied by manufacturers of fruit and cereal bars to improve their health-promoting properties included: 1/enrichment with essential fatty acids; 2/enrichment with vitamins and minerals; 3/reduction in fat content; 4/reduction in sugar content; 5/reduction in salt content; 6/reduction in cholesterol content; 7/enrichment with protein; 8/removal of allergy-inducing ingredients; 9/enrichment with fibre; 10/addition of insect powder; 11/addition of fruit; 12/addition of vegetables; 13/addition of chia seeds/flaxseed, insects; 14/addition of nuts. Respondents provided their opinions towards each expected modification on a 5-point scale: 1—“the modification matters very little to me” and 5—“the modification matters very much to me”. There was no neutral point on this scale. A 5-point ordinal scale was used, in which only the two extreme poles are described.

To determine respondents’ opinions on what information on the packaging/label of an innovative fruit and cereal bar with health-promoting properties could encourage consumers to buy it, the following information was taken into account: 1/contains no sugar, 2/contains only natural sugars derived from fruit, 3/increased fibre content, 4/helps maintain healthy body weight, 5/fibre helps digestion, 6/natural source of fibre, 7/a celebrity image on the packaging, 8/costs less, 9/more product for the same price. Respondents were given the opportunity to choose the following answers: yes, no, I have no opinion. 

### 2.4. Statistical Analysis

As a preliminary analysis of the results obtained, a characterization of the variables was performed. Descriptive statistics were used for the quantitative variables and frequency analysis for the qualitative variables. Based on this, it was decided to create a logistic regression model, in which the dependent variable was the respondents’ interest in an innovative fruit and cereal bar on the packaging of which the manufacturer declared its health-promoting properties. Due to the qualitative nature of this characteristic, a logistic regression model with a dichotomous (binary) dependent variable was used. The terms: interest in purchasing an innovative fruit and cereal bar, familiarity with innovative fruit and cereal bars, expectations of changes that could be applied by fruit and cereal bar manufacturers to improve their health-promoting properties, and expectations of information provided on the packaging of an innovative fruit and cereal bar about health-promoting properties were used as independent (explanatory) variables. Only statistically significant explanatory variables were used in the model created.

The C-statistic value and the Hosmer and Lemeshow goodness-of-fit test were used to assess the quality of the resulting model. All analyses were performed using the SAS 9.4 statistical package (SAS Institute, Cary, NC, USA) at a significance level of α = 0.05 [25,26].

## 3. Results

### 3.1. Characteristics of the Survey Sample

The socio-demographic characteristics, including gender, age, place of residence, level of education, and subjective evaluation of financial situation, are presented in Table 1.

Approximately 70% of the respondents were women. More than half of the respondents were aged 25–39 years. The smallest number of respondents were over 55 years of age. Most of the respondents lived in a city. More than 50% of the respondents had a university education, and about 40% had a secondary education. More than half of the respondents could afford some, but not all, expenses. For about 15%, income was insufficient and allowed them to meet only basic needs.

### 3.2. Interest in Purchasing and Information on an Innovative Fruit and Cereal Bar with Health-Promoting Properties

More than half of the respondents (53.6%) declared that the innovative fruit and cereal bar arouses their interest, but that making a purchase decision requires thinking about it, gathering information, and comparing it with other products in the category.

Almost one in three respondents (35.4%) want to try it out and, if possible, buy without consideration. Only 7.0% of respondents embrace any novelty, including a fruit and cereal bar, with distrust, and, if they do make a purchase, it is only after their acquaintances have purchased the product and are satisfied with it. Around 4% of respondents declared that innovative products generally do not interest them and they do not buy untested products.

Among the changes made by producers to fruit and cereal bars, respondents perceived mainly changes to organic packaging, the use of new preservation methods, the enrichment of products with new ingredients, and the reduction in the level of ingredients unfavourable to health. In contrast, the least frequently perceived changes were changes to the product weight, packaging appearance, and size (Table 2).

Among the expected innovations in health-promoting fruit and cereal bars, the reduction in sugar content and the addition of fruit were indicated, in addition to the addition of vitamins and minerals, insect powder, and chia seeds/flaxseed. The least expected was the addition of nuts.

For more than four-fifths of the respondents, the image of a celebrity on the packaging/label of a fruit and cereal bar with health-promoting properties was an incentive to buy it. In addition, respondents indicated information such as “helps maintain healthy body weight”, “natural source of fibre”, and “more product for the same price”. By far the fewest people (23.1%) perceived the information “contains only natural sugars derived from fruit” as encouraging the purchase of the product (Table 3).

### 3.3. Determinants of Purchasing an Innovative Fruit and Cereal Bar with Health-Promoting Properties

Respondents declaring their willingness to try an innovative fruit and cereal bar and buying it without thinking about it were up to 25 times more likely to be interested in buying a fruit and cereal bar with health-promoting properties compared with the reference level, i.e., people who generally do not buy new, untested products (OR: 25.2, 95% CI: 8.82–71.9). In contrast, for those in whom an innovative bar arouses interest, but they consider whether to buy it, gather information, compare it with others, and only then buy it, the chance was more than 17 times higher relative to the reference level (OR: 17.4, 95% CI: 6.37–47.9). Those declaring that they accept any novelty with distrust, take longer to think about buying it, and only buy a product when their friends have already done so, showed an over 3.5 times greater chance of being interested in purchasing a fruit and cereal bar with health-promoting properties relative to the reference level (OR: 3.66, 95% CI: 1.21–11.0) (Table 4).

The more the respondents were convinced that the innovative actions of fruit and cereal bar manufacturers consisted of changing their packaging to eco-friendly and enriching the product with new ingredients, the more likely they were to purchase an innovative bar with health-promoting properties, increasing by 23% (OR: 1.23, 95% CI: 1.03–1.48) and 34% (OR: 1.34, 95% CI: 1.06–1.69), respectively. In contrast, perceptions of changes as involving the introduction of new ways of preserving food (no preservatives, no colourings) decreased the chance of purchasing such a product by 26% for every one-point increase in this opinion (OR: 0.74, 95% CI: 0.59–0.94); see Table 4.

Among the changes expected by respondents that increased the chances of purchasing a fruit and cereal bar with health-promoting properties were: 1/addition of flaxseed or chia seeds—an increase of 22% (OR: 1.22, 95% CI: 1.10–1.42); 2/enrichment of the bar with vitamins and minerals—an increase of 19% (OR: 1.19, 95% CI: 1.10–1.42); 3/removal of allergy-inducing ingredients—an increase of 19% (OR: 1.19, 95% CI: 1.07–1.36); 4/addition of vegetables—an increase of 13% (OR: 1.13, 95% CI: 1.09–1.24); 5/protein enrichment—an increase of 11% (OR: 1.11, 95% CI: 1.04–1.31); see Table 4. 

The expected changes in fruit and cereal bars that reduced the chances of purchasing such a health-promoting product were: 1/reduction in sugar content—6% decrease (OR: 0.94, 95% CI: 0.79–0.98); 2/reduction in cholesterol content—11% decrease (OR: 0.89, 95% CI: 0.74–0.96); and 3/enrichment with insect powder—27% decrease (OR: 0.73, 95% CI: 0.61–0.89); see Table 4. 

The presence of a claim on the packaging/label of a health-promoting fruit and cereal bar indicating that the product “Helps maintain a healthy body weight” increased the chance of purchasing this product by 51% (OR: 1.51, 95% CI: 1.06–2.45) compared with those who did not agree with this opinion. When an image of a celebrity was placed on the product packaging, the chance of purchasing an innovative fruit and cereal bar with health-promoting properties increased almost twofold in the group of people accepting it (OR: 1.86, 95% CI: 1.02–3.40). Those who perceived the information “Natural source of fibre” on the packaging/label of the innovative bar as encouraging the purchase of the bar were 37% less likely to purchase a fruit and cereal bar with health-promoting properties compared with those opposed to this opinion (OR: 0.63, 95% CI: 0.36–0.97) (Table 4). All assessed odds ratios (ORs) refer to when the other variables in the model remain constant.

## 4. Discussion

The study aimed to ascertain consumers’ interest in innovative fruit and cereal bars and their expectations of changes that could be applied by manufacturers to improve their health-promoting properties. In addition, the aim was to determine how these interests and expectations, as well as the information provided on the product label, determine the willingness to purchase a fruit and cereal bar with health-promoting properties.

Research suggests that commercially available cereal bars are characterised by varying nutritional value [12]; among other things, they may contain high levels of sugar and saturated fat [27]. The presence of such products on the market may cause them to be perceived as ultra-processed foods, with high amounts of additives, and thus as unnatural products [21]. A study of the German snack bar market found that although consumers categorised this product as highly processed, of 812 snack bars (protein, cereal, fruit, and nuts), 734 were bars that were not nutritionally poor and unnatural [28]. It is noted that their consumption is increasing worldwide [2,3]. As consumers become more health-conscious, cereal bars have gradually gone from a “standard” product to a “custom-made” product by integrating different functional ingredients [15,29]. The snack industry keeps investing to find more healthy innovative alternatives or substitutes to design bars [30].

The results of our survey confirmed that the willingness to purchase a fruit and cereal bar with health-promoting properties was mainly declared by those who showed general interest in fruit and cereal bars but were also characterised by innovation towards this type of product. Nevertheless, opinions regarding current and expected changes to these products and the nature of the information on their labels also determined the likelihood of interest in purchasing such bars. Opinions indicating the need for the addition of flaxseed or chia seeds, vitamins, minerals, and protein, and the removal of allergy-inducing ingredients, increased interest in purchasing fruit and cereal bars with health-promoting properties. In contrast, expectations of lowering the sugar and cholesterol content of these products, as well as the addition of powdered insects, were linked to a lower interest in purchasing such products. 

The survey found that respondents expected future changes to fruit and cereal bars that would increase their health-promoting properties, including a reduction in sugar and added fruit, vitamins, and minerals, but also insect powder and chia seeds/flaxseed. The expectation of consumers to improve the health properties of fruit and cereal bars by reformulating the fruit and cereal bars already available on the market is confirmed by the actions of their manufacturers. They enrich of these products with whole grains; nuts [31]; seeds—flaxseed, quinoa, amaranth, chia [32]; vegetables—Welsh onion [33]; fruits—strawberry, raspberry, cranberry, jackfruit, dates, apple, banana peel [31,34]; legumes—lentil, beans, soybeans, Bambara groundnut [35]. Healthy ingredients rich in vitamins, minerals, amino acids, omega-3, and bioactive compounds are used to improve the nutritional value of the products and positively impact health [36,37]. As noted by [38], bioactive compounds of plant origin (polyphenols, antioxidants, bioactive peptides, and probiotics) are gaining consumer acceptance and may be health-promoting ingredients in cereal bars in the future. 

A study by [39] confirmed that high levels of consumer acceptance could be gained for bars produced with the addition of flaxseed and quinoa, brown rice, nuts, and honey. The addition of vegetables and legumes as sources of fibre, minerals, antioxidants, and proteins rich in essential amino acids is also accepted [30]. Positive attitudes towards various additives that increase the nutritional value of a fruit and cereal bar were also declared by the respondents in our study, as they were positive about changes involving the addition of flaxseed and chia seeds, vitamins and minerals, but also about the addition of vegetables to fruit and cereal bars with health-promoting properties. Such changes, respondents declared, also prompted them to purchase innovative bars with health-promoting properties.

Our study showed that the consumers were interested in the addition of protein to the fruit and cereal bar, which also increased the chance of purchasing it. This result confirms one of the emerging market trends associated with cereal bars, which is protein fortification [30]. The increased interest in physical activity, including both endurance and strength sports, requires an increase in protein intake [33]. For this reason, protein cereal bars are gaining popularity, but also because of the importance of protein in weight and appetite control, satiety, and reducing daily food intake, especially among conscious consumers [40]. Currently, proteins from peas, lentils, lupins, algae [41], and insects [42,43,44] are used in the production of cereal bars in addition to traditional protein sources from soy or dairy products. The latter was also indicated by respondents as an expected change to the fruit and cereal bars, although the logistic regression shows that this type of innovation decreases the chance of purchasing such a product. Thus, it can be presumed that there is an acceptance of adding powdered insects to bars, but at the same time, consumers are quite sceptical about this type of action. Research confirms that the rejection of edible insects is due to the social and cultural norms that characterise different areas of the world [42]. In Poland, as in other European countries, cultural conditions may cause consumer aversion to this new food category [45,46,47]. In some regions of the world, which include primarily Latin America, Asia, and Africa, the consumption of edible insects is a common practice. This is related to the habit of consuming them, the tradition of consumption in the region, easy access, or low price [48]. On the one hand, edible insects are characterised by their favourable chemical composition and high nutritional value [49,50], but on the other hand, the susceptibility of fat extracted, for example, from the mealworm to rapid oxidation, may favour the formation of compounds with unpleasant taste and smell and thus reduce the acceptability of the products [51].

Less interest in purchasing a fruit and cereal bar with health-promoting properties was found when the sugar content of the product was reduced, even though such a modification was one of the most expected by respondents. These results indicate that although respondents declared that they expected such changes, they were not ready to purchase such products, perhaps because they still associate bars with a “sweet snack” and, as indicated by [52], this may be due to consumers being used to the sweet taste and not being willing to change their habits. It is noted that the snack industry, meeting the expectations of consumers as well as doctors and nutritionists, is looking for innovative alternatives to the sugar-reduced cereal bar. In addition, manufacturers are also aiming to reduce fat and sodium in these products [38] as well as gluten [12,39]. 

In contrast, according to respondents, the chance of purchasing an innovative fruit and cereal bar with health-promoting properties would be higher if reformulations were to reduce the content of unfavourable allergy-inducing ingredients. The use of gluten-free flour may be important due to the increasing number of patients with celiac disease [53]. Hence, the use of pseudo-cereals, e.g., grain amaranth, quinoa, buckwheat, chenopods, and chia seeds, is becoming more common [54].

The results of the study allow us to conclude that respondents show a high level of interest in innovative fruit and cereal bars, which may be related to the relatively high demand for functional foods among consumers [55]. 

Respondents who perceive that the bars are enriched with new ingredients are more likely to purchase an innovative bar with health-promoting properties. To create an innovative fruit and cereal bar, manufacturers are enriching them with new functional ingredients or lowering the level of ingredients that are unfavourable for health, which is confirmed by many studies [15,30]. The outcome of these efforts is the availability of many types of bars on the market. These include standard or fortified (e.g., fruits, pseudo-cereals, pulses, and insects); reduced in fat or sugar; gluten-containing or gluten-free; laminated or extruded; and single, multilayer, or sandwich format [56]. 

Furthermore, many studies confirm consumer interest in bars with cereals as the main ingredient [29,30]. They are consumed on the go, as meal replacements, and during exercise as products that are increasingly in consumers’ minds as positively influencing health and associated with natural foods [15,57]. The positive impact of consumed cereal bars on consumer health can be evidenced by the results of a study confirming that the addition of fibre-rich quinoa resulted in a reduction in total cholesterol, low-density lipoprotein (LDL) cholesterol, and triglycerides in young people eating cereal bars [37]. In addition, research shows that eating a cereal bar before noon instead of other snacks (e.g., crisps, sweets, biscuits, and cakes) can improve mental well-being, good mood, and memory (mental health) [58]. 

Respondents indicated that, currently, manufacturers of fruit and cereal bars increasing their innovativeness primarily focus on changing their packaging to eco-friendly. Further efforts by manufacturers in this area would be an important predictor in the decision to purchase an innovative fruit and cereal bar with health-promoting properties. Many cereal bar manufacturers are adopting plastic-free packaging and using renewable plant-based materials [30,59]. 

The results of our survey confirmed that consumers declare that they use the information provided on the packaging when deciding to purchase an innovative fruit and cereal bar. One of the functions of product packaging is to highlight those characteristics of the goods that are most desirable to the consumer. Hence, various forms of information about the product’s unique characteristics are used, e.g., nutrition and health claims and graphic symbols to accentuate product features that may be important to the purchaser [8,60]. Our study found that the willingness to purchase an innovative fruit and cereal bar with health-promoting properties would be higher if its packaging stated that it “helps to maintain a healthy body weight”. Normal body weight contributes to maintaining good health and feelings of being satisfied with life [61]. It is most often obtained through physical activity and proper nutrition [62]. It is noted that any action taken that promotes a healthy lifestyle, also related to the choice of health-promoting products, can reduce the risk of overweight and obesity and the development of related chronic diseases, such as diabetes [63]. 

According to the respondents, the chance of purchasing the product category under study would also be higher if an image of a celebrity was placed on the packaging. The commercial potential of the image of popular figures from the world of show business, politics, and sport has also been recognised by other researchers [64]. The sympathy and respect enjoyed by well-known and talented individuals cause consumers to attribute those characteristics to products associated with their idols. According to research, in recent years, food influencers, through social media, have proven to be very effective in endorsing health-promoting foods [65]. Information from them is perceived as more trustworthy and more reliable compared with other sources of information [66]. In addition, because of their relationship with their followers, these individuals can influence consumers’ opinions and purchase intentions [67]. 

As in the case of other food products [68,69], an incentive for consumers of fruit and cereal bars to purchase them can also be provided by information about their naturalness [70] and increased fibre content [36]. The results of our study, on the other hand, show that if manufacturers included information on the packaging of an innovative health-promoting fruit and cereal bar that it is a “natural source of fibre”, the chance of purchasing it would be 37% lower. Consideration could be given to including a statement on the packaging of health-promoting fruit and cereal bars that they are made from natural ingredients, that they are organic, or that they are free from ingredients that are not beneficial to health. This suggestion is based, among other things, on the results of a survey of US consumers, half of whom bought cereal bars on the packaging of which the following information was included: “made with natural ingredients”, “organic”, “free-from” [71]. 

## 5. Strength and Limitations

A strength of the study was a large group of people who declared that they consume fruit and cereal bars. Such a group allowed us to determine the current and projected preferences for an innovative fruit and cereal bar with health-promoting properties. This information can be used by manufacturers to prepare the bar in line with consumer expectations, which may result in a higher level of acceptance of this product when it appears on the market as well as fostering a more pro-healthy diet for different categories of consumers for whom snacking between meals is popular; it is better to use snacks with health-promoting qualities rather than salty or sugary ones. 

Due to the lack of national statistics on the consumption of fruit and cereal bars, a limitation of the study was the non-random sampling. As a result of such selection, the group was dominated by women. In addition, a limitation of the study was that it was conducted only among Polish residents, although many of the results obtained are also confirmed by worldwide reports. There is a need to undertake further research detailing consumer preferences for a fruit and cereal bar with health-promoting properties in other countries.

## 6. Conclusions

In the context of the presented results of the quantitative survey on a nationwide research sample, it can be concluded that consumers are relatively innovative and interested in a snack in the form of a fruit and cereal bar with health-promoting properties. Nevertheless, it is important for manufacturers to introduce an addition to their composition or to reduce the content of certain ingredients that are unfavourable for health reasons, e.g., sugar.

It was found that manufacturers’ actions to enrich fruit and cereal bars with chia seeds or flaxseed, vitamins and minerals, as well as vegetables and protein, and to remove ingredients that cause allergies, would statistically significantly increase the chance of purchasing such bars. A celebrity image and a claim that the product “helps to maintain a healthy body weight” on the label would also increase the willingness to purchase it. On the other hand, reducing the sugar content or enriching a fruit and cereal bar with powdered insects would significantly reduce the propensity to buy it. It was also found that changing the packaging to biodegradable and therefore an environmentally friendly one would also increase interest in purchasing this type of product. 

In addition to its cognitive value, the collected empirical material also has an important application aspect, indicating to producers the directions for reformulation of fruit and cereal bars and ways of communicating their health-promoting qualities.

## Figures and Tables

**Table 1 nutrients-16-01606-t001:** Socio-demographic characteristics of the study sample.

Variables		Total Sample	
		N	%
Total Sample		1034	100.0
Gender	Woman	716	69.3
	Man	318	30.7
Age	18–24 years	145	14.0
	25–39 years	556	53.8
	40–54 years	283	27.4
	55–65 years	50	4.8
Place of residence	Rural	185	17.9
	City up to 100,000 inhabitants	377	36.5
	City with more than 100,000 inhabitants	472	45.6
Education	Primary education	16	1.6
	Basic vocational	76	7.4
	Secondary	411	39.7
	Higher	531	51.3
Opinion on income	Is not sufficient at all	26	2.5
	Allows us to meet only basic needs	127	12.3
	We can afford some, but not all, expenses	572	55.3
	We can afford everything	227	22.0
	We can afford everything, plus we can save	82	7.9

**Table 2 nutrients-16-01606-t002:** Respondents’ opinions on changes made and expected changes to fruit and cereal bars.

Statements	Mean Value	Standard Deviation	Modal Value
Current innovative efforts undertaken by fruit and cereal bar manufacturers *			
Changing to organic packaging	3.92	1.15	4
Introducing new ways of preserving food	3.86	1.10	4
Enriching the product with new ingredients	3.80	1.05	4
Reducing levels of components that are detrimental to health	3.78	1.09	4
Changing the product composition	3.69	1.05	4
Implementing changes to protect the environment	3.64	1.12	3
Introducing a product with a new flavour	3.36	1.17	3
Improving packaging to improve product shelf life	3.32	1.13	3
Suggesting new uses for the product	3.24	1.17	3
Changing the appearance of the packaging	2.94	1.19	3
Changing the size of the packaging	2.91	1.19	3
Changing the product weight	2.78	1.21	3
Innovative changes expected by consumers in a health-promoting fruit and cereal bar **			
Reduction in sugar content	4.07	1.08	5
Addition of fruit	4.05	0.96	5
Enrichment with vitamins and minerals	3.96	1.01	5
Addition of powdered insect powder	3.96	1.04	5
Addition of chia seeds/flaxseed	3.92	1.05	5
Reduction in cholesterol	3.90	1.10	5
Enrichment with protein	3.89	1.10	5
Removal of ingredients causing allergies	3.88	1.09	5
Enrichment with fibre	3.72	1.09	3
Reduction in salt content	3.71	1.11	4
Reduction in fat content	3.63	1.24	3
Addition of vegetables	3.62	1.16	3
Enrichment with essential fatty acids	3.52	1.11	3
Addition of nuts	3.18	1.16	3

* Scale: 1 indicating “totally disagree”, 3 indicating “neutral”, and a score of 5 indicating “totally agree”. ** Scale: 1, “the modification matters very little to me” and 5, “the modification matters very much to me”.

**Table 3 nutrients-16-01606-t003:** Information on the packaging of a fruit and cereal bar encouraging purchase as perceived by respondents (%).

Information on the Packaging	Information as an Incentive to Purchase		
	Yes	No	I Have No Opinion
It does not contain sugar	62.1	20.9	17.0
It contains only natural sugars derived from fruit	23.1	58.0	18.9
Increased fibre content	67.9	16.0	16.1
It helps maintain a healthy body weight	75.4	12.3	12.3
Fibre helps digestion	65.9	18.2	15.9
Natural source of fibre	71.7	14.8	13.5
Showing an image of a celebrity on the packaging	80.7	8.5	10.8
Costs less	63.0	20.6	16.4
More product for the same price	70.2	16.3	13.5

**Table 4 nutrients-16-01606-t004:** Statistically significant variables and their estimation properties used to build the logistic regression model.

Variable		Estimate	Point Estimate	Pr > ChiSq
Consumer interest in an innovative fruit and cereal bar appearing in shops		−3.089 (−2.478)		0.0003 (0.0286)
	I want to try it out and if it’s possible, I buy it right away without a second thought	3.227 (3.166)	25.201 (23.716)	<0.0001 (<0.0001)
	It arouses my interest, I wonder if I will buy it, I collect information, I compare it with others and finally, I buy it.	2.861 (2.803)	17.478 (16.489)	<0.0001 (<0.0001)
	I accept any novelty with distrust, think long and only buy when my friends have already done so and are happy with their purchase	1.298 (1.248)	3.660 (3.485)	0.0214 (0.0282)
	New products generally do not interest me, I do not buy untested things	0	1	
Current innovative efforts undertaken by producers of a fruit and cereal bar with health-promoting properties	Changing the packaging to eco-friendly	0.214 (0.208)	1.238 (1.231)	0.0207 (0.0266)
	Enriching the product with new ingredients	0.294 (0.304)	1.341 (1.355)	0.0145 (0.0126)
	Implementing new ways of preserving food	−0.291 (−0.296)	0.747 (0.744)	0.0127 (0.0125)
Innovative changes expected by consumers in a health-promoting fruit and cereal bar	Enriching the product with vitamins and minerals	0.176 (0.153)	1.192 (1.165)	0.0398 (0.0412)
	Reducing sugar content	−0.061 (−0.038)	0.941 (0.963)	0.0493 (0.0573)
	Reducing cholesterol	−0.116 (−0.121)	0.891 (0.886)	0.0319 (0.4179)
	Enriching the product with protein	0.112 (0.104)	1.119 (1.110)	0.0173 (0.0221)
	Removing allergy-inducing ingredients	0.175 (0.167)	1.191 (1.182)	0.0125 (0.0188)
	Addition of powdered insects	−0.304 (−0.320)	0.738 (0.726)	0.0020 (0.0014)
	Addition of fruit	0.130 (0.076)	1.138 (1.079)	0.0255 (0.0429)
	Addition of chia seeds/flaxseed	0.202 (0.228)	1.224 (1.256)	0.0152 (0.0076)
Information on the packaging/label “Helps maintain a healthy body weight”	I have no opinion	0.091 (0.067)	1.095 (1.069)	0.7789 (0.8394)
	Yes	0.415 (0.395)	1.515 (1.485)	0.0399 (0.0311)
	No	0	1	
Information on packaging/label “Natural source of fibre”	I have no opinion	−0.112 (−0.068)	0.894 (0.935)	0.7740 (0.8640)
	Yes	−0.450 (−0.417)	0.637 (0.659)	0.0446 (0.0462)
	No	0	1	
Showing an image of a celebrity on the packaging/label	I have no opinion	0.007 (0.004)	1.007 (1.004)	0.9805 (0.9887)
	Yes	0.624 (0.641)	1.866 (1.899)	0.0418 (0.0386)
	No	0	1	

The correctness of the prediction of the model created was confirmed by the statistics C = 0.838 and Hosmer and Lemeshow goodness-of-fit test (*p* = 0.8474) for model crude (unadjusted) and C = 0.840 and Hosmer and Lemeshow goodness-of-fit test (*p* = 0.3706) for model adjusted used to assess it.

## Data Availability

The data is property of the Warsaw University of Life Sciences—SGGW.

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
