# Peer review of "Consumer Perception of Innovative Fruit and Cereal Bars—Current and Future Perspectives"

_nutrients, 2024, doi:10.3390/nu16111606_

Round 1

Reviewer 1 Report

Comments and Suggestions for Authors

Interesting manuscript that aims: to ascertain consumers' interest in innovative fruit and cereal bars and their expectations of changes that could be applied by manufacturers to improve their health-promoting properties.

I have no comments on the introduction or the results, but I do have doubts in the discussion:

1. cereal bars can be classified as an ultra-processed food, and a growing number of health professionals are against them, can there be healthy ultra-processed foods and if there is literature to support your favorable comment.

2. among the low expectations of consuming a cereal bar, was to reduce the sweet taste content, did they not explore with the substitution of polyols, tagatose, allulose or non-caloric sweeteners? what does the literature say about it?

3. in the substitution of insects, issues of food culture have been observed, for example, in countries where insects are usually consumed, a cereal bar with insect flour does not produce rejection or this rejection is independent of the country? please explain 

Author Response

Reviewer 1

Dear Reviewer,

Thank you very much for your very valuable comments and for the time you dedicated to reviewing our article.

We have highlighted our changes in yellow in the manuscript.

The following are our responses:

  1. cereal bars can be classified as an ultra-processed food, and a growing number of health professionals are against them, can there be healthy ultra-processed foods and if there is literature to support your favorable comment.

Research suggests that commercially available cereal bars are characterised by varying nutritional value (Aleksejeva et al., 2017), among other things, they may contain high levels of sugar and saturated fat (New Zealand Ministry of Health, 2020). The presence of such products on the market may cause them to be perceived as highly processed, with a high amount of additives, and thus as unnatural products (Perkovic et al., 2021). Furthermore, they are snacks eaten between main meals. However, it is noted that their consumption is increasing worldwide (Nordman et al., 2020; Djupegot et al., 2021). They are consumed by people on a diet, those involved in sports, people with health problems or consumers looking to satisfy hunger quickly (Farinazzi-Machado et al., 2012).

In the last decade, consumers have become more concerned about health and well-being and are paying more attention to the food that they eat (Klerks et al., 2022). As consumers become more health-conscious, cereal bars have gradually gone from a “standard” product to a “custom-made” product by integrating different functional ingredients (Pallavi et al., 2015; Perez-Rodríguez et al., 2023). The snack industry keeps investing to find innovative alternatives or substitutes to design bars with reduced content of some nutrients such as sugar, fat and sodium (Boukid et al., 2022).

Research furthermore shows that cereal bars can be a healthier alternative to chocolate bars. They can be a lower-calorie dietary supplement (Klerks et al., 2022). The enrichment of bars with nuts, fruits and cereals promotes their acceptance due to their bioactive content (Bucher et al., 2016; Huitink et al., 2020; Poquet et al., 2020). Studies indicate that their consumption is also determined by respondents' awareness that they provide more fibre (Aleksejeva et al., 2017; Curtain & Grafenauer, 2019). In addition, healthy ingredients rich in vitamins, minerals, amino acids, omega-3, and bioactive compounds are used to formulate cereal bars with a high nutritional value in response to various but specific target groups (Farinazzi-Machado et al. 2012).

We agree with you and we pointed out in the Introduction that despite the beneficial properties of cereal bars mentioned above, researchers emphasise that they belong to the category of sweets and cannot be considered products with exclusively positive health effects (Pallavi et al. 2015). Cereal bars might have some drawbacks as well, such as the free sugar added to the formulation of the binder to act as a sticky agent in the product’s assembly. There is evidence that high intakes of added and free sugars increase the risk of developing chronic metabolic diseases including obesity, non-alcoholic fatty liver disease, type 2 diabetes, dyslipidaemia and hypertension, possibly through an increase in energy intake and body weight, among other mechanisms (Klerks et al., 2022).

In Discussion, we added: that commercially available cereal bars are characterised by varying nutritional value. Among other things, they may contain high levels of sugar and saturated fat.

Only an educated choice on the part of consumers who read the information on the packaging will allow them to choose from the wide range of cereal bars on the market, which can have a positive impact on health.

References:

Aleksejeva, S., Siksna, I., & Rinkule, S. (2017). Composition of cereal bars. Journal of Health Science, 5(3), 139-145.

Boukid F., Klerks M., Pellegrini N., Fogliano V., Sanchez-Siles L., Roman S., Vittadini E. (2022). Current and emerging trends in cereal snack bars: implications for new product development. International Journal of Food Sciences and Nutrition, 73(5), 610-629.

Brito A.L.B., Brito L.R., Honorato F.A., Pontes M.J.C., Pontes L.F.B.L. (2013). Classification of cereal bars using near infrared spectroscopy and linear discriminant analysis. Food research international, 51(2), 924-928.

Bucher T., Collins C., Diem, S., Siegrist M. (2016). Adolescents’ perception of the healthiness of snacks. Food Quality and Preference, 50, 94–101.

Curtain F., Grafenauer S. (2019). Comprehensive nutrition review of grain-based muesli bars in Australia: An audit of supermarket products. Foods, 8(9), 1–13.

Djupegot I.L., Hansen S., Lähteenmäki L. (2021). What you say and what you do: exploring the link between consumers’ perception of portion size norms and reported behaviour for consumption of sweets and crisps. Food Quality and Preference, 92, 104216.

Farinazzi-Machado F.M.V., Barbalho S.M., Oshiiwa M., Goulart R., Pessan Junior O. (2012). Use of cereal bars with quinoa (Chenopodium quinoa W.) to reduce risk factors related to cardiovascular diseases. Food Science and Technology, 32, 239-244.

Huitink M., Poelman M.P., Seidell J.C., Pleus M., Hofkamp T., Kuin C., Dijkstra S.C. (2020). Can unhealthy food purchases at checkout counters be discouraged by introducing healthier snacks? A real-life experiment in supermarkets in deprived urban areas in the Netherlands. BMC Public Health, 20(1), 542.

Klerks M., Román S., Verkerk R., Sanchez-Siles L. (2022). Are cereal bars significantly healthier and more natural than chocolate bars? A preliminary assessment in the German market. Journal of Functional Foods, 89, 104940.

New Zealand Ministry of Health. (2020). Eating and Activity Guidelines. In Ministry of Health – Manat¯u Hauora. http://www.health.govt.nz/our-work/eating-and-activity-guidelines.

Nordman, M., Matthiessen, J., Biltoft-Jensen, A., Ritz, C., & Hjorth, M. (2020). Weekly varioation in diet and physical acitivity among 4–75-year-old Danes. Public Health Nutrition, 23(8), 1350–1361.

Pallavi, B. V., Chetana, R., Ravi, R., & Reddy, S. Y. (2015). Moisture sorption curves of fruit and nut cereal bar prepared with sugar and sugar substitutes. Journal of Food Science and Technology, 52(3), 1663–1669.

Pérez-Rodríguez M., Hidalgo M.J., Mendoza A., González L.T., Rodríguez F.L., Goicoechea H.C., & Pellerano R.G. (2023). Measuring trace element fingerprinting for cereal bar authentication based on type and principal ingredient. Food Chemistry: X, 18, 100744.

Perkovic, S., Otterbring, T., Sch¨arli, C., & Pachur, T. (2021). The Perception of Food Products in Adolescents, Lay Adults, and Experts: A Psychometric Approach. Journal of Experimental Psychology: Applied.

Poquet, D., Ginon, E., Goubel, B., Chabanet, C., Marette, S., Issanchou, S., & Monnery-Patris, S. (2019). Impact of a front-of-pack nutritional traffic-light label on the nutritional quality and the hedonic value of mid-afternoon snacks chosen by mother-child dyads. Appetite, 143, 104425.

  1. among the low expectations of consuming a cereal bar, was to reduce the sweet taste content, did they not explore with the substitution of polyols, tagatose, allulose or non-caloric sweeteners? what does the literature say about it?

We have confirmed that among the low expectations of consuming a cereal bar, was to reduce the sweet taste content. We have not researched consumer expectations of sugar substitution of polyols, tagatose, allulose or non-caloric sweeteners.

Increasing cases of obesity-related to sugar intake have led to a greater need for studies with sugar substitutes (Mooradian et. al., 2017).

Sugar (as a binder) can be replaced by other ingredients, even though finding the right balance between technological, sensorial and nutritional quality is very challenging (Boukid et al., 2022).

Sugars, as well as syrups (e.g. dextrose syrup, sucrose, maltodextrin, inverted sugar syrup, dextrose, and fructose), are widely used as binders and sweeteners, but they also act as improvers of product stability during storage (due to the water binding ability of amorphous sugars) (Wang and Ryu 2013; Farahnaky et al., 2016). When present in an amorphous status, they also confer to the bar a chewy and flexible texture. The major drawback of these ingredients is related to their negative effects of increasing glycaemia (Pallavi et al., 2015). Alternative glueing agents (i.e. fibres and polyols) with low glycemic response can be used to promote the binding effect and to substitute sugar-based syrups (Pallavi et al., 2015; Srebernich et al., 2016). Polyols are nutritive and natural sweeteners. They are low-digestible carbohydrates, which reduce calories and can be used as a sugar substitute. Polyols consumed in large quantities can cause a laxative effect (Grembecka 2015).

In most cases, commercial products are made from mixtures of intense sweeteners and polyols. However, new innovative low-calorie sugar substitutes are more relevant to the development of cereal bars than sweeteners because sweeteners can perform one function of sugar (adding sweetness) but cannot provide a binding effect. Psicose, also known as allulose, is a promising new innovative sugar substitute (Mooradian et al. 2017).

Concluded that sugar reduction negatively affects the texture of bars often resulting in a hard product. To overcome this problem, in some cases, recipes have been adapted by adding fat and/or glycerine, testing different combinations of syrups or making changes to the processing (Di Monaco et al. 2018).

References:

Boukid F., Klerks M., Pellegrini N., Fogliano V., Sanchez-Siles L., Roman S., Vittadini E. (2022). Current and emerging trends in cereal snack bars: implications for new product development. International Journal of Food Sciences and Nutrition, 73(5), 610-629.

Di Monaco R., Miele N.A., Cabisidan E.K., Cavella S. (2018). Strategies to reduce sugars in food. Current Opinion in Food Science. 19:92–97.

Farahnaky A, Mansoori N, Majzoobi M, Badii F. (2016). Physicochemical and sorption isotherm properties of date syrup powder: antiplasticizing effect of maltodextrin. Food Bioprod Process. 98:133–141.

Grembecka M. (2015). Sugar alcohols—their role in the modern world of sweeteners: a review. European Food Research and Technology. 241(1):1–14.

Mooradian A.D., Smith M., Tokuda M. (2017). The role of artificial and natural sweeteners in reducing the consumption of table sugar: a narrative review. Clinical Nutrition ESPEN Journal. 18:1–8.

Pallavi BV, Chetana R, Ravi R, Reddy SY. (2015). Moisture sorption curves of fruit and nut cereal bar prepared with sugar and sugar substitutes. J Food Sci Technol. 52(3):1663–1669.

Srebernich S.M., Goncalves G.M.S., Ormenese R.d.C.S.C., Ruffi C.R.G. (2016). Physico-chemical, sensory and nutritional characteristics of cereal bars with addition of acacia gum, inulin and sorbitol. Food Sci Technol. 36(3): 555–562.

Wang Y-Y, Ryu G-H. (2013). Physicochemical and antioxidant properties of extruded corn grits with corn fiber by CO2 injection extrusion process. Journal of Cereal Sciences, 58(1): 110–116.

  1. in the substitution of insects, issues of food culture have been observed, for example, in countries where insects are usually consumed, a cereal bar with insect flour does not produce rejection or this rejection is independent of the country? please explain.

In some regions of the world, which may include primarily Latin America, Asia and Africa, the consumption of edible insects is a common practice. This is related to the habit of consuming them, the tradition of consumption in the region, easy access, or low price (Raheem et al., 2019).

In the face of growing threats of a future global food crisis, edible insects are a viable source of food for humans and feed for animals (high nutritional value) (Ordonez-Araque et al., 2022), and they are a suitable form of responsible consumption. In this regard, the interest in edible insects has grown over time (Valesi et al., 2024). They have also started to be recognized as a novel food source in Western countries (Lange and Nakamura, 2021). However, despite these recent advances, acceptance of edible insects by Western consumers is still very low (Mancini et al., 2021). Many research investigations have been conducted to understand why consumers reject edible insects and which variables can positively influence their acceptance. Most research has focused on psychological traits and consumer characteristics, identifying the high sensitivity of consumers coming from Western Europe to disgust at insects and food neophobia as the main predictors of entomophagy rejection. These products for this part of the world are unfamiliar and still difficult to accept (Cunha and Ribeiro, 2019). Research confirms that the rejection of edible insects is due to the social and cultural norms that characterise different areas of the world (Ribeiro et al., 2022).

Research also shows that one way to get consumers used to edible insects is not to offer them in a whole piece, but to add insect powder to products commonly accepted by consumers. These products include, but are not limited to: protein/energy bars, protein shakes, and bakery products/snacks (Ardoin and Prinyawiwatkul, 2020). These products with the addition of powdered edible insects, for people who are not used to their taste, are often unacceptable. Confirmation of this finding can be found in several studies (Cunha and Ribeiro, 2019; Ribeiro et al., 2019), including one by researchers from Portugal (Ribeiro et al., 2022). Five different formulations of oats dehydrated and dried fruit bars were formulated. The different formulations consisted of a control bar – CTRL – (without incorporation of insect) and four formulations incorporating insects (at a 15% level): defatted mealworm (TM_D), whole ground microwave-dried (TM_MW) or electrical oven dried (TM_O) mealworm and whole ground cricket dried in an electrical oven (AD_O). Results from overall liking and willingness to eat, show that species and processing technologies had a significant effect. All the bars that incorporated insects had lower liking and acceptance than a control bar.

References:

Ardoin R., Prinyawiwatkul W. (2020). Product appropriateness, willingness to try and perceived risks of foods containing insect protein powder: A survey of US consumers. International Journal of Food Science & Technology, 55(9), 3215-3226.

Cunha L. M. Ribeiro J. C. (2019). Sensory and consumer perspectives on edible insects. Edible insects in the food sector: Methods, current applications and perspectives, 57-71.

Lange K.W., Nakamura Y. (2021). Edible insects as future food: chances and challenges. Journal of Future Foods 1. 38–46.

Mancini S., Mattioli S., Paolucci S., Fratini F., Dal Bosco A., Tuccinardi T., & Paci G. (2021). Effect of cooking techniques on the in vitro protein digestibility, fatty acid profile, and oxidative status of mealworms (Tenebrio molitor). Frontiers in Veterinary Science. 8, 675572.

Ordonez-Araque, R., Quishpillo-Miranda, N., & Ramos-Guerrero, L. (2022). Edibleinsects for humans and animals: Nutritional composition and an option for mitigating environmental damage. Insects. 13(10), 1–13

Raheem D., Carrascosa C., Oluwole O.B., Nieuwland M., Saraiva A., Millán R., Raposo A. (2019). Traditional consumption of and rearing edible insects in Africa, Asia and Europe. Critical Reviews in Food Science and Nutrition. 59, 2169–2188.

Ribeiro J. C., Santos C., Lima R.C., Pintado M.E., & Cunha L.M. (2022). Impact of defatting and drying methods on the overall liking and sensory profile of a cereal bar incorporating edible insect species. Future Foods, 6, 100190.

Ribeiro J.C., Lima R.C., Maia M.R.G., Almeida A.A., Fonseca A.J.M., Cabrita A.R.J., Cunha L.M. (2019). Impact of defatting freeze-dried edible crickets (Acheta domesticus and gryllodes sigillatus) on the nutritive value, overall liking and sensory profile of cereal bars. LWT 113, 108335.

Valesi R., Andreini D., & Pedeliento G. (2024). Insect-based food consumption: Hedonic or utilitarian motives? Moderation and segmentation analyses. Food Quality and Preference, 105193.

Thank you very much for all your comments. We appreciate them very much. We hope that we were able to improve the article according to your suggestions. Thank you for taking the time to read and evaluate the text.

Reviewer 2 Report

Comments and Suggestions for Authors

Title, Abstract, and Keywords:

The title relates well with the described work, and is brief but elucidative.

The abstract is well organized but lacks some quantitative results. Just an  example when saying “It was found that producers’ efforts to change the packaging to an organic one along with enriching the product with chia seeds/flaxseed, vitamins and minerals as well as vegetables and protein, or removing ingredients that cause allergies would significantly increase the  chance  of  purchasing  such  a  bar. “, it woyld be advisable to quantify the actual increase for each of the factors presented.

Introduction:

The introduction helps to frame and contextualize the work. It presents some state of the art on a number of topics which are essential to the work that was carried out, and it serves as a justification for its development.

Materials and methods:

The description of the methodologies applied to obtain and treat the data are presented clearly, in general.

In line 117, maybe you could add information about the middle point of the scale? It usually is neither agree nor disagree, but it must be specified. This seems like a symmetrical scale, (negative / positive, with a NEUTRAL middle point)

In relation  to scale in line 127, this is even more complex, because this scale, although having also 5 points, is all on the positive side “matters very little…” until “matters very much…”, which means that the middle point is not Neutral!!!! In fact in this scale when you respond 1, it matters, even if very little it is a positive side.

Although apparently both scales are similar, varying from 1 to 5, they are in fact very different in nature, and it must be explained what the different points mean, specifically the middle point – score = 3).

Results and discussion:

The presentation of results is clear in general, in the form of Tables.

In Table 2 it must be added a footnote explaining the scale for mean values, since it is relevant to understand what the values are in relation to the scale. Remember that there are two different scales, so you must refer to them both, and also remember to indicate what they mean, including specifically the extreme points and the middle point of each scale.

Discussion:

The discussion is thorough and the authors use a number of references to support their discussion and correlate with other published works.

Conclusions and limitations

The conclusions part is also well enough, presenting the most relevant findings of the work. Some limitations were pointed out, which are useful to understand the extent of the conclusions of the study.

References

There is an acceptable rate of recent references, so the authors should make an effort to have a high number off references from the last 5 years.

Comments on the Quality of English Language

English is generaly fine

Author Response

Reviewer 2

Dear Reviewer,

Thank you very much for your very valuable comments and for the time you dedicated to reviewing our article.

We have highlighted our responses in green in the manuscript.

Title, Abstract, and Keywords:

The title relates well with the described work, and is brief but elucidative.

The abstract is well organized but lacks some quantitative results. Just an  example when saying “It was found that producers’ efforts to change the packaging to an organic one along with enriching the product with chia seeds/flaxseed, vitamins and minerals as well as vegetables and protein, or removing ingredients that cause allergies would significantly increase the  chance  of  purchasing  such  a  bar. “, it woyld be advisable to quantify the actual increase for each of the factors presented.

The values of the indicators next to the factors have been added.

Introduction:

The introduction helps to frame and contextualize the work. It presents some state of the art on a number of topics which are essential to the work that was carried out, and it serves as a justification for its development.

Materials and methods:

The description of the methodologies applied to obtain and treat the data are presented clearly, in general.

In line 117, maybe you could add information about the middle point of the scale? It usually is neither agree nor disagree, but it must be specified. This seems like a symmetrical scale, (negative / positive, with a NEUTRAL middle point)

As suggested the explanation has been added :“neither agree nor disagree”.

In relation  to scale in line 127, this is even more complex, because this scale, although having also 5 points, is all on the positive side “matters very little…” until “matters very much…”, which means that the middle point is not Neutral!!!! In fact in this scale when you respond 1, it matters, even if very little it is a positive side.

Although apparently both scales are similar, varying from 1 to 5, they are in fact very different in nature, and it must be explained what the different points mean, specifically the middle point – score = 3).

There is no neutral point on this scale. A 5-point ordinal scale was used, in which only the two extreme poles were described. The lowest rating of 1 meant - - ‘the modification matters very little to me’ and the highest rating of 5 meant ‘the modification matters very much to me’. This is a positive scale. Despite the differences between the scales, for both scales, the higher the value, the more positive the attitude towards the statement.

As suggested the explanation has been added.

Results and discussion:.

The presentation of results is clear in general, in the form of Tables.

In Table 2 it must be added a footnote explaining the scale for mean values, since it is relevant to understand what the values are in relation to the scale. Remember that there are two different scales, so you must refer to them both, and also remember to indicate what they mean, including specifically the extreme points and the middle point of each scale.

Thank you for your comments. As suggested the explanation has been added.

Discussion:

The discussion is thorough and the authors use a number of references to support their discussion and correlate with other published works.

Conclusions and limitations

The conclusions part is also well enough, presenting the most relevant findings of the work. Some limitations were pointed out, which are useful to understand the extent of the conclusions of the study.

References      

There is an acceptable rate of recent references, so the authors should make an effort to have a high number of references from the last 5 years.

We have completed the manuscript a high number off references from the last 5 years.

Comments on the Quality of English Language

The article has been checked for correct use of the English language by the co-authors. In addition, we have asked native speakers to check the correctness of English language usage.

Thank you very much for all your comments. We appreciate them very much. We hope that we were able to improve the article according to your suggestions. Thank you for taking the time to read and evaluate the text.

Round 2

Reviewer 1 Report

Comments and Suggestions for Authors

The authors made the requested changes satisfactorily, the manuscript is accepted.